# Seizures in PPT1 Knock-In Mice Are Associated with Inflammatory Activation of Microglia

**DOI:** 10.3390/ijms23105586

**Published:** 2022-05-17

**Authors:** Xusheng Zhang, Mengting Wang, Bingyan Feng, Qiuyu Zhang, Jia Tong, Mingyong Wang, Chengbiao Lu, Shiyong Peng

**Affiliations:** 1Institute of Psychiatry and Neuroscience, Xinxiang Medical University, Xinxiang 453003, China; zxsarticle@126.com (X.Z.); mengting2022@126.com (M.W.); zhangqiuyu678@126.com (Q.Z.); shenyangyaodatj@hotmail.com (J.T.); 2Henan International Joint Laboratory of Non-Invasive Neuromodulation, Department of Physiology and Pathophysiology, Xinxiang Medical University, Xinxiang 453003, China; 18839191970@163.com; 3Xinxiang Key Laboratory of Immunoregulation and Molecular Diagnostics, School of Laboratory Medicine, Xinxiang Medical University, Xinxiang 453003, China; wmy118@126.com

**Keywords:** seizures, microglia, A 438079, PPT1 KI mice, hippocampus

## Abstract

Infantile neuronal ceroid lipofuscinosis (INCL), the most severe form of neuronal ceroid lipofuscinoses, is caused by mutations in the lysosomal enzyme palmitoyl protein thioesterase 1 (PPT1). Typical symptoms of this disease include progressive psychomotor developmental retardation, visual failure, seizures, and premature death. Here, we investigated seizure activity and relevant pathological changes in PPT1 knock-in mice (PPT1 KI). The behavior studies in this study demonstrated that PPT1 KI mice had no significant seizure activity until 7 months of age, and local field potentials also displayed epileptiform activity at the same age. The expression levels of Iba-1 and CD68 demonstrated, by Western blot analysis, the inflammatory cytokine TNF-α content measured with enzyme-linked immunosorbent assay, and the number of microglia demonstrated by immunohistochemistry (IHC) were significantly increased at age of 7 months, all of which indicate microglia activation at an age of seizure onset. The increased expression of GFAP were seen at an earlier age of 4 months, and such an increase reached its peak at age of 6 months, indicating that astrocyte activation precedes microglia. The purinergic P2X7 receptor (P2X7R) is an ATP-sensitive ionic channel that is highly expressed in microglia and is fundamental to microglial activation, proliferation, cytokines release and epilepsy. We show that the ATP concentration in hippocampal tissue in PPT1 KI mice was increased using an enhanced ATP assay kit and demonstrated that the antagonist of P2X7R, A-438079, significantly reduced seizures in PPT1 KI mice. In contrast to glial cell activation and proliferation, a significant reduction in synaptic proteins GABA_A_R was seen in PPT1 KI mice. These results indicate that seizure in PPT1 KI mice may be associated with microglial activation involved in ATP-sensitive P2X7R signaling and impaired inhibitory neurotransmission.

## 1. Introduction

Neuronal ceroid lipofuscinosis (NCL) is a group of autosomal recessive neurodegenerative diseases characterized by the accumulation of ceroid lipofuscin deposition in the brain [1]. Common clinical symptoms of the disease include impairment of cognitive and motor functions, blindness, refractory seizures, and premature death [2,3,4,5]. Infantile neuronal ceroid lipofuscinosis (INCL) is the fastest developing and most severe type of NCL [6] caused by mutations in the *CLN1* gene, which encodes palmityl protein thioesterase-1 (PPT1). As a depalmitoylase, PPT1 plays a critical role in regulating protein transport and protein–protein interactions. The mutation or deletion of PPT1 can induce a series of neurological pathological changes, including decreased synaptic plasticity, the activation of glial cells, and neural death [7,8,9].

Mice, as a common rodent model, were used to simulate INCL through engineered knock-out of relevant genes or knock-in mutated genes [10,11]. The homozygous PPT1 knock-out mouse (PPT1 KO) with deletion of exon 4 or 9 of the *PPT1* gene showed pathological symptoms similar to those of the human INCL, while deletion mutations were not genetically analogous to any human *CLN1* mutations. The most common *CLN1* mutation is the *R151X* mutation, accounting for 52.3 of INCL cases [12]. Therefore, knock-in mouse models have been created by introducing the *R151X* mutation into exon 5 of the mouse *PPT1* gene [12,13]. Previous work has demonstrated that PPT1 KI mice developed lipofuscin storage materials throughout the brain, displaying INCL-like pathological injury, neural apoptosis, gliosis activation, and retinal degeneration.

Neuronal network oscillations at γ frequency band (30–80 Hz, γ oscillation) emerge from networks of mutually connected interneurons and pyramidal neurons, which provide a time frame for the synchronization of firing of the neurons within the network, critical for information processing [14], and play an important role in many higher brain functions, such as learning and memory [15,16]. Hippocampal γ oscillations reflect the physiological functions of GABAergic interneurons from the local network, and interneuron dysfunction leads to impairment of γ oscillations and/or epileptiform discharge [17,18].

Seizure is a common symptom of many neurological diseases, which is characterized by the abnormal hypersynchrony of neuronal activity in the brain [19]. The pathogenesis of seizures is the increase in excitatory neurotransmitter and/or the decrease in inhibitory neurotransmitters [20,21]. Previous studies on seizures have focused on neural injury, activation of glial cells, and release of inflammatory cytokines, such as interleukin (IL)-1β, tumor necrosis factor-α (TNF-α), and nitric oxide [22,23,24], which mediate inflammatory cell death [25,26]. INCL patients have seizure onset at a young age, and frequent seizures can be also seen in PPT1 KO mice at 3 months old [27], but Kielar et al. (2007) did not observe seizures in PPT1 KO mice until 7 months of age [7]. The seizures and related mechanisms in PPT1 KI mice have not been reported.

Microglia are critical immune cells of the central nervous system. They play a key role in physiological functions and pathological changes in the brain [28]. Microglia are highly implicated in the etiology of neurological diseases, such as Alzheimer’s disease and epilepsy [29,30]. The purinergic P2X7 receptor (P2X7R), a member of the P2X receptor family, is an adenosine triphosphate (ATP)-gated cation channel that is mainly expressed in microglia, as well as astrocytes, oligodendrocytes and neurons at a lower density [31,32]. P2X7R has been reported to be critically involved in seizures [33,34]. Activated by ATP that is released from neuronal terminals, leaked from the damaged cellular membrane of neurons, or outpoured from astrocytes [35], P2X7 mediates inflammatory activation of microglia. Activated microglia release inflammatory cytokines, which results in neuronal inflammation, damage, and seizure onset [36,37,38,39]. P2X7R has been considered a potential target for the treatment of epilepsy.

Kielar et al. [7] demonstrated that activated microglia were significantly increased in the primary motor cortex, somatosensory cortex, and primary visual cortex with increased age in PPT1-KO mice; however, time-dependent changes in microglial activity have not been reported in PPT1 KI mice. Here, we attempted to investigate the possible mechanism underlying seizures in PPT1 KI mice and the relationship between microglial activation and seizure occurrence.

## 2. Results

### 2.1. Seizures in PPT1 KI Mice

To observe seizures in PPT1 KI mice, we recorded daily activities of PPT1 KI mice (5–7 months old) and WT mice (7 months old) for 24 h. PPT1 KI mice had no seizures at 5–6 months but exhibited typical seizures (over stage 4) at 7 months of age (Figure 1A). The observation of seizure praxeology showed that seizure levels and duration varied largely in PPT1 KI mice. Seizure levels were mostly scaled as stage 2 (head nodding) and stage 3 (forelimb clonus), with few at stage 4 (rearing) and stage 5 (rearing and falling) (Figure 1B). The longest seizure duration was up to 10 min 57 s, and the shortest was less than 1 min. These results demonstrate that PPT1 KI mice could suffer from seizures, as seen in PPT1 KO mice, and that these seizures occurred in late age in PPT1 KI mice.

### 2.2. Epileptiform Activity in Brain Slices of PPT1 KI Mice

To investigate the function of neural network in PPT1 KI mice, a low concentration of kainic acid (KA, 200 nM) was applied to the hippocampal slices, and γ oscillations in hippocampal CA3 from the WT mice were induced, as reported previously [40], but there was epileptiform activity from PPT1 KI mice at 7 months of age (Figure 2A,D). The typical examples of epileptiform activity in these slices are characterized from field potential recordings by the burst firing of bi-directional complex waves (0.1–0.4 Hz), composed of a large negative potential (0.5–2 mV, wave width 20–200 ms) followed by a large positive potential (0.2–2 mV, 50–300 ms) (Figure 2D (d1) and (d2)). The γ oscillation in WT mice showed a stable increase over time, indicating a normal local neural network mediated by inhibitory interneurons (Figure 2B,C). In PPT1 KI mice, the epileptiform activity emerged 5 min after KA treatment, and the number of epileptiform activities reached stability at 10 min (Figure 2E). These results suggest the impaired inhibition and neural network dysfunction in the hippocampus of 7-month-old PPT1 KI mice.

### 2.3. Microglial Activation in Hippocampus Correlates with the Occurrence of Seizures

To investigate the expression level of Iba-1, a microglial marker, in PPT1 KI mice, we performed Western blot analysis. The results showed that the expression level of Iba-1 was significantly increased in the hippocampus of 7-month-old PPT1 KI mice compared with age-matched WT mice (*p* = 0.04, permutation *t* test, *n* = 6) (Figure 3A).

To further demonstrate the correlation between microglial activation and seizures in PPT1 KI mice, the number of microglia in the hippocampus of 1–7-month-old PPT1 KI mice was measured. The number of hippocampal microglia did not change in PPT1 KI mice from age 1 to 6 months, but it was significantly increased at 7 months of age (*p* < 0.001, one-way ANOVA on ranks followed by post hoc LSD test, *n* = 35) (Figure 3B). The expression level of Iba-1 in the hippocampus of WT mice had no change from age 1 to 7 months (Figure 3C). We also analyzed expression levels of CD68, a marker of activated microglia and macrophages. The expression levels of CD68 in PPT1 KI mice showed a slow and small increase from age 1 to 6 months and a large increase at 7 months of age (1 vs. 4 months, *p* = 0.049, 6 vs. 7 months, *p* = 0.03, one-way ANOVA on ranks followed by post hoc LSD test, *n* = 21) (Figure 3D). There were no significant changes in the expression level of CD68 in WT mice from age 1 to 7 months (Figure 3E). These results indicate that the number of activated microglia was significantly increased in 7-month-old PPT1 KI mice.

### 2.4. Astrocyte Is Activated at the Early Stage of PPT1 KI Mice

The expression of the astrocyte marker GFAP was determined by Western blot analysis. The level of GFAP was significantly elevated in PPT1 KI mice starting at 4 months of age and continued to increase with increased age until 6 months old (1 vs. 4 months, *p* = 0.01; 6 vs. 7 months, *p* = 0.625, one-way ANOVA on ranks followed by post hoc LSD test, *n* = 21 in Appendix A), indicating that increased astrocyte activation occurs at the early stage of PPT1 KI mice.

### 2.5. Age-Dependent Changes in Synaptic Proteins GluN2B and GABA_A_Rα1 in PPT1 KI Mice

The expression of excitatory synaptic protein, the GluN2B subunit of N-methyl d-aspartate receptors (NMDARs), and inhibitory synaptic protein GABA_A_Rα1 were further determined by Western blot analysis. GluN2B-containing NMDARs are expressed in hippocampal interneurons, and GABA_A_R mediates inhibitory synaptic transmission [41,42]. Our results showed the decreased expression of GluN2B in the hippocampus of PPT1KI mouse (GluN2B, 1 vs. 7 months, *p* < 0.01, one-way ANOVA on ranks followed by post hoc LSD test, *n* = 35 in Appendix A) and a similar pattern of decreased expression of hippocampal GABA_A_Rα1 in PPT1KI mouse from age 1 to 7 months (GABA_A_Rα1, 1 vs. 7 months, *p* < 0.01, one-way ANOVA on ranks followed by post hoc LSD test, *n* = 35) (Figure 3F), which indicates the impaired inhibitory synaptic transmission in the hippocampus of PPT1 KI mice with age. There was no age-related change in the expression of GABA_A_Rα1 in WT mice (Figure 3G).

### 2.6. Iba-1 Immunoreactivity and Morphological Changes of Microglia in Hippocampus of PPT1 KI Mice

To further determine the relationship between seizures and microglial activation in PPT1 KI mice, we performed in situ IHC in the hippocampus of 7-month-old PPT1 KI mice and age-matched WT mice (Figure 4A). The microglia of the WT mice remained in a resting state with slender branches, whereas the microglia had larger somata with shorter, thicker and dense protrusions in the hippocampus of 7-month-old PPT1 KI mice, indicating that these microglia are highly activated. Our results showed that the number of microglia in the hippocampal CA1 and CA3 of PPT1 KI mice was significantly increased compared to those of age-matched WT mice (CA1: *p* = 0.003, CA3: *p* = 0.01, permutation *t* test, *n* = 12) (Figure 4B,C). Our data further showed that microglial activation was significantly increased in PPT1 mice with seizures at 7 months of age, which suggests that activated microglia likely contributed to seizures in the PPT1 KI mice.

### 2.7. Age-Related Neuronal Loss in PPT1 KI Mice

To test whether there is neuronal loss in PPT1 KI mice, we used DAB staining IHC in the hippocampus of PPT1 KI mice at ages 3, 5 and 7 months. Our data demonstrated that there was no significant neuronal loss in 3- and 5-month-old PPT1 KI mice, but there was 46% neuronal loss in 7-month-old PPT1 KI mice compared to 5-month-old PPT1 KI mice (5 vs. 7 months, *p* < 0.001, one-way ANOVA on ranks followed by post hoc LSD test, *n* = 12) and 7-month-old WT mice (*p* < 0.001, Student’s *t* test, *n* = 12) (Figure 4D,E). Therefore, these results indicate that PPT1 KI mice with seizure are associated with dramatic neuronal death and microglial activation.

**Figure 4 ijms-23-05586-f004:**
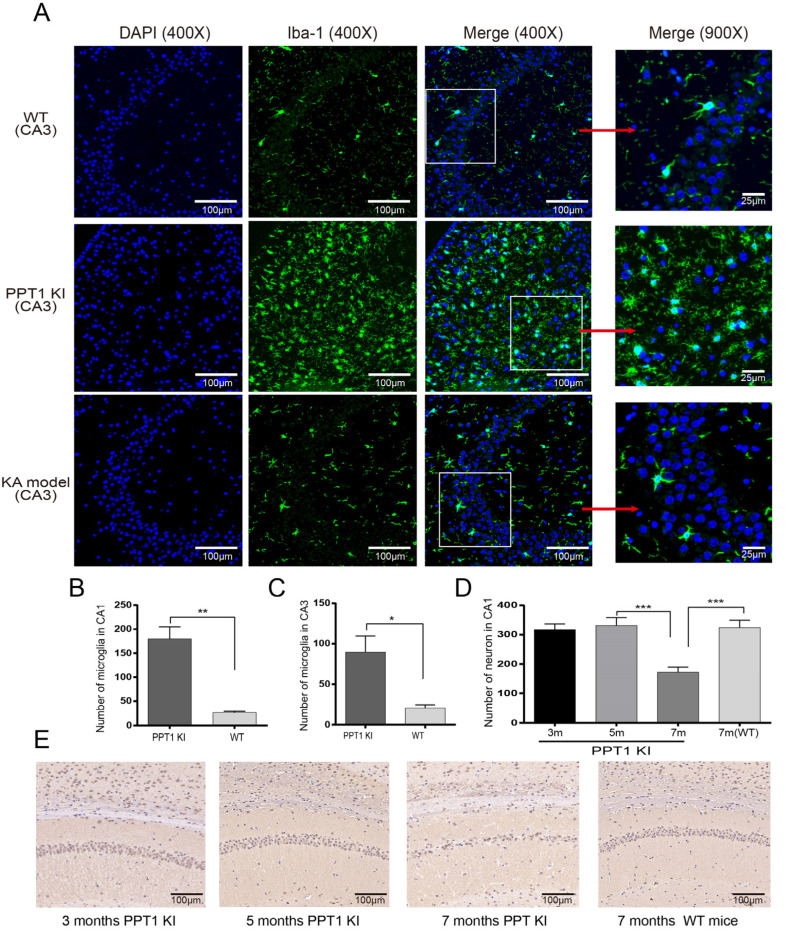
Immunoreactivity of Iba-1 in the hippocampus of 7-month-old WT, PPT1 KI and KA model mice. (**A**) Representative images of confocal imaging in hippocampal CA3 area of 7-month-old WT, PPT1 KI and KA model mice. The panels in the left two columns show the images of hippocampal slices treated with DAPI (blue) and Iba-1 antibody (green fluorescence) acquired at 400× magnification. The square regions marked by white lines within the merged image (400×) were selected and expanded to show morphological changes of microglia at 900× magnification (right panels). (**B**) The number of microglia in the hippocampal CA1 of PPT1 KI mice and WT mice (*p* = 0.003, permutation *t* test, *n* = 12). (**C**) The number of microglia in the hippocampal CA3 of PPT1 KI mice and WT mice (*p* = 0.005, permutation *t* test, *n* = 12). (**D**) The number of neuronal cells in hippocampal CA1 of 3-, 5- and 7-month-old PPT1 KI mice and 7-month-old WT mice (5 vs. 7 months, *p* < 0.001, one-way ANOVA on ranks followed by post hoc LSD test, 7-month-old PPT1 KI mice vs. 7-month-old WT mice, *p* < 0.001, Student’s *t* test, *n* = 12). (**E**) Representative images of DAB staining in hippocampal CA1 at 200× magnification. * *p* < 0.05; ** *p* < 0.01; ****p* < 0.001.

### 2.8. Cytokines IL-1β and TNF-α Changes in PPT1 KI Mice

We measured the contents of two common cytokines IL-1β and TNF-α by ELISA. Our results showed that the concentrations of IL-1β slightly increased in PPT1 KI mice compared with WT, but there was no statistically significant difference (*p* = 0.1, permutation *t* test, *n* = 12) (Figure 5A). The expression levels of TNF-α in PPT1 KI mice were significantly increased compared to their WT littermates (WT vs. PPT1 KI, *p* = 0.02, permutation *t* test, *n* = 10) (Figure 5B).

### 2.9. Inhibition of ATP-Sensitive P2X7R Repressed Seizure in PPT1 KI Mice

We measured the concentration of ATP in the hippocampus of 7-month-old PPT1 KI mice (seizure state and resting state) and age-matched WT mice. The ATP concentration of PPT1 KI mice was significantly higher in the seizure state than the resting state (*p* = 0.043, permutation *t* test, *n* = 6) (Figure 6A). The ATP concentration in the resting state of PPT1 KI mice was higher than that of WT mice (*p* = 0.048, permutation *t* test, *n* = 6) (Figure 6A).

Intraperitoneal injection of A 438079 (30 mg/kg), an ATP-sensitive P2X7 antagonist, into 7-month-old PPT1 KI mice significantly reduced the duration and number of seizures (seizure duration, *p* = 0.01, number of seizures, *p* = 0.01, Permutation *t* test, *n* = 8). There was no significant change in the saline (30 mg/kg) treatment group (Figure 6B,C). Critically, A 438079 treatment significantly reduced the number of microglia in hippocampal CA1 and CA3 of PPT1 KI mice (CA1, *p* = 0.004, CA3, *p* = 0.02, permutation *t* test, *n* = 12) (Figure 6D–F). These data indicate that the P2X7R inhibitor alleviated seizures, suggesting the possible microglia involvement in the seizures of PPT1 KI mice via P2X7R activation.

## 3. Discussion

In this study, behavioral experiments showed that PPT1 KI mice had no seizures until 7 months of age. The behavioral finding was confirmed by ex vivo LFP recording in PPT1 KI mouse. Western blot and immunoreactivity experiments demonstrated microglia activation and increased ATP concentration in PPT1 KI mice with seizure. Furthermore, inhibition of P2X7R, an inotropic ATP receptor mainly expressed in microglia, reduced the number and total duration of seizures in PPT1 KI mice.

Seizures and epileptiform activities observed in 7-month-old PPT1 KI mice both in vivo and ex vivo indicate the impaired hippocampal neural networks in 7-month-old PPT1 KI mice. A foundational mechanism of seizures is imbalance between excitatory and inhibitory neurotransmission. Previous studies on the pathogenesis of seizures in PPT1 KO mice were attributed to the loss of GABAergic interneurons [7]. The PPT1 KI mice had motor deficits during 3–5 months of age [12], but seizure onset was at 7 months of age. The reason for this late-onset seizure is probably related to the sufficient expression level of the GABA_A_Rα1 maintained in PPT1 KI mice until 5 months of age. While the PPT1 KI mice exhibit behavioral seizures at 7 months of age, it is possible that the mice experience electrographic seizures that manifested as behavioral seizures only later [43].

Some studies have suggested that astrocytes are involved in epilepsy [44]. The astrocytes of PPT1 KO mice showed functional and morphological abnormalities, and their survival rate was lower than that of WT mice [45]. Our results showed that the number of astrocytes significantly increased even at an early age (3 months) in PPT1 KI mice, which is in line with previous reports [46,47]. Although it is not consistent with the timing of seizure onset in these mice, the early and dramatic increase in GFAP expression in PPT1 KI mice in this study and other reports [46] suggest that astrocytes play a role in neuroinflammation and microglial activation in PPT1 KI mice in response to neuronal damage and death in these mice.

Dramatic neuronal death was observed in the hippocampus in PPT1 KI mice by our immunoreactivity data. The mechanisms for neuronal death in PPT1 KI mice have been studied, and the abnormalities in NMDARs in PPT1 KI mice appear to be involved [42]. The increased expression of GluN2B in neurons of PPT1 KO mice may render neurons vulnerable to excitotoxicity [48]. Our data showed that the expression levels of GluN2B were not significantly altered in PPT1 KI mice until age 6–7 months, the reason for the downregulation of GluN2B at such a late age is currently unknown but may be related to the reduced neuronal numbers in these mice. Nevertheless, the increased inflammatory cytokines such as TNF-α measured in this study are known to mediate neuronal death [25,26]. The reduced expression of GABA_A_Rα1 becomes evident at the late stage of PPT1 KI mice, suggesting that the reduced inhibition mediated by GABA _A_Rα1 may also contribute to increased excitotoxicity, seizure, and neuronal loss in these mice.

Although there is existing neuronal loss in PPT1 KI mice, this study demonstrated that the number of activated microglia was significantly increased in PPT1 KI mice with seizures, which is in agreement with previously published reports [12,49]. The morphological changes in microglia of PPT1 KI mice were consistent with the microglial activation observed in the hippocampus of WT mice with seizures induced by KA [50]. In humans, the expression of microglial markers in the hippocampus is significantly higher in a human brain with seizures than in normal individuals [51]. The duration and frequency of seizures are correlated with microglial activation [49].

Critically, unlike in astrocyte activation, which precedes seizure, microglial activation is concurrent with seizure onset in PPT1 KI mice, emphasizing a role of microglial activation in seizure onset. The binding of P2X7R located on microglial membrane to ATP, released from astrocytes and/or neurons, is an important mechanism to activate microglia and release microglial cytokines (IL-1β and TNF-α) [36]. The inflammatory cytokine IL-1β expression was increased in the brains of epileptic mice [52,53] and in stimulated PPT1 KO microglia [45]. These cytokines act on neurons and astrocyte to release ATP and glutamate, and consequently cause neuronal hyperexcitation, excitotoxicity and seizures [54,55] (Figure 7).

The involvement of inflammatory cytokine and microglia in seizures was also demonstrated by the evidence that the inhibition of cytokine production attenuates neuroinflammation and seizures [56,57]. Application of PLX3397, a potent inhibitor of colony stimulating factor-1 receptor expressed on the cell surface of microglia depletes microglial activation, by which neuronal injury and seizure activity were reduced and clinical outcome was improved in PPT1 KO mice [54], which both demonstrate a detrimental impact on the microglia in the CNS of *CLN1* mice.

In our study, the increased expression of cytokines TNF-α and an increasing trend of IL-1β expression may support the increased inflammation and neuronal injury being associated with microglial/astrocyte activation in PPT1 KI mice.

Our data further showed that the ATP concentration significantly increased in the hippocampal tissue of PPT1 KI mice in the seizure state, suggesting that astrocytes and neurons in these mice are under stress and release a large amount of ATP [58,59]. The increased ATP likely acts on microglia and neurons in PPT1 KI mice. Therefore, the interference in the role of ATP may be effective for the treatment of INCL. In this study, administration of A 438079, a P2X7R antagonist to PPT1 KI mice, significantly reduced the number and total duration of seizures. This result was in agreement with a previous report that P2X7R inhibitors reduce microgliosis and seizures in kainic acid-induced epileptic mice [33,60,61].

In summary, we provide evidence to demonstrate that PPT1 KI mice have seizures, which is associated with inflammatory activation of microglia involved in ATP-P2X7R signaling.

## 4. Materials and Methods

### 4.1. Animals

Homozygous mutant mice of PPT1 c.451C > T/c.451C > T, termed PPT1 KI, were generously gifted by our collaborator, Dr. Anil B. Mukherjee, Eunice Kennedy Shriver National Institute of Child Health and Human Development, National Institutes of Health (Bethesda, MD, USA) (Appendix A). Mice were housed at Xinxiang Medical University Laboratory Animal Center and bred in a pathogen–free facility with 12 h light/12 h dark cycles with access to water and food ad libitum. There were 74 PPT1 KI mice and 21 age-matched wild-type (WT) littermates, C57BL6, used in this study. The smallest possible number of mice was used in the experiments according to sample size test (MSST v. 6.0.6). All animal procedures were performed according to guidelines approved by the committee on animal care at Xinxiang Medical University.

### 4.2. Behavioral Scaling in Seizures

PPT1 KI mice aged from 5–7 months old and 7-month-old WT mice were placed into the observation cage with water and food at constant room temperature (25 ± 2 °C). Mice were videotaped by the Plexon CinePLex Studio Application Version 3.7 (Plexon, Dallas, TX, USA) for 24 h in 12 h/12 h day/night cycle (lights on at 8 a.m.). The spontaneous seizure behavior repertoire of mice was classified into five stages according to the Racine scale [62], including: (1) mouth and facial movements, (2) head nodding, (3) forelimb clonus, (4) rearing, and (5) rearing and falling. We recorded the duration and number of seizures within 24 h.

### 4.3. Slice Preparation and Extracellular Field Recordings

KI and WT mice were anesthetized by intraperitoneal injection of pentobarbital (40 mg/kg). The brains were removed and stored in a cold slicing solution (225 mM sucrose, 3 mM KCl, 6 mM MgSO_4_; 1.2 mM NaH_2_PO_4_; 24 mM NaHCO_3_, 10 mM glucose, and 0.5 mM CaCl_2_). For local field potential (LFP) recordings, 350 µm horizontal sections were made using a Leica VT1000S vibratome (Leica Microsystems, Milton Keynes, England, UK). The hippocampal slices were transferred to an interface recording chamber and incubated for 1 h. Slices were maintained at a temperature of 32 °C and at the interface between the artificial cerebrospinal fluid (ACSF) and warm humidified carbon gas (95% O_2_–5% CO_2_). The ACSF contained (in mM): 126 NaCl; 3 KCl; 2 MgSO_4_; 1.25 NaH_2_PO_4_; 24 NaHCO_3_; 10 glucose, and 2 CaCl_2_. Extracellular field recordings were performed using the interface-recording chambers. Recordings were obtained using normal ACSF-filled 1–3 MΩ borosilicate glass microelectrodes (Sutter Instrument, Novato, CA, USA). The γ oscillation from hippocampal CA3 was induced through the perfusion of ACSF containing kainic acid (200 nM). Data were band-pass filtered online between 0.5 Hz and 2 kHz using the Axoprobe amplifier and a Neurolog system NL106 AC/DC amplifier (Digitimer Ltd. Cambridge, England, UK). The data were digitized at a sample rate of 5–10 kHz using a CED 1401 plus ADC board (Digitimer Ltd. Cambridge, England, UK). Electrical interference from the main supply was eliminated from extracellular recordings online using 50 Hz noise eliminators (HumBug; Digitimer Ltd. Sequim, WA, USA).

Data were analyzed offline using Spike 2 software (CED, Cambridge, UK). Power spectra were generated to provide a quantitative measure of the frequency components in a stretch of recording, where power, a quantitative measure of the oscillation strength, was plotted against the respective frequency. Power spectra were constructed for 60 s epochs of extracellular field recordings using a fast Fourier transform algorithm provided by Spike 2. The parameters used for measuring the oscillatory activity in the slice were the peak frequency (Hz) and area power (μV2). In this study, the area power was equivalent to the computed area under the power spectrum between the frequencies of 20 and 60 Hz.

### 4.4. Western Blotting

The hippocampal tissues from mice were homogenized with 200 μL RIPA (R0020, Solarbio, Beijing, China) and 2 μL protease inhibitor (4693132001, Roche, Mannheim, Germany, 2 ml H_2_O per tablet). The homogenates were on the shaker for 30–40 min in 4 °C, followed with centrifugation (3000 rpm, 10 min, 4 °C). Total protein concentration in the supernatant was determined with Bicinchoninic Acid assay (P0009, Beyotime Biotechnology, Shanghai, China). For Western blotting, protein samples were resolved by electrophoresis using 5–12% sodium dodecyl sulfate–polyacrylamide gels under denaturing and reducing conditions and electrotransferred to nitrocellulose membranes (pore size 0.2 μm, PALL, 66485). The membranes were blocked with 5% nonfat dry milk (BD, 232100) and then subjected to the immunoblot analysis using standard methods. The primary antibodies used for the immunoblots were as follows: Iba-1 (A19776, 1:1000, ABclonal), CD68 (E-AB-40066,1:1000, Elabscience), glial fibrillary acidic protein (GFAP; 60190-1-Ig, 1:5000, Proteintech); GABA_A_Rα1 (ab252430, 1:1000, Abcam); GluN2B (21920-1-AP, 1:1000, Proteintech), and β-actin (ab8226, 1:500, Abcam) as a loading control. The blot was incubated with the primary antibodies overnight at 4 °C in Tris-buffered saline-Tween-20. After 3 washes in Tris-buffered saline-Tween-20, blots were with horseradish peroxidase (HRP) goat anti-rabbit immunoglobulin G (IgG) (AS014, 1:5000, ABclonal) and HRP goat anti-mouse IgG (AS003, 1:5000, ABclonal) at room temperature for 1 h. Thereafter, immunoreactivity was detected by chemiluminescence (Tanon, 180-501, Shanghai, China). The membranes were scanned using a Tanon scanner (1600R), and the results were analyzed using AlphaEaseFC 4.0.0 (Alpha Innotech).

### 4.5. Enzyme-Linked Immunosorbent Assay (ELISA)

The protein concentrations of IL-1β and TNF-α were quantified using an ELISA kit (4A Biotech Co., Ltd., Beijing, China) according to the manufacturer’s instruction for 7-month-old PPT1 KI mice and age-matched WT mice. The optical density was detected at a wavelength of 450 nm, and the concentration of the target protein was calculated according to the standard curve and normalized against the protein of the samples. Results were expressed as pg/mg protein. Sensitivity: the lowest detectable IL-1β and TNF-α dose is less than 15 pg/mL. Both intra- and interassay CV% are less than 10%.

### 4.6. ATP Assays

The unilateral hippocampus was harvested from PPT1 KI mice with seizures and without seizures and age-matched WT control mice. The ATP concentrations of these hippocampi were assessed using an enhanced ATP assay kit (Beyotime, Shanghai, China) according to the manufacturer’s protocol. ATP consumption was calculated using the ATP standard curve.

### 4.7. Drug Administration

The specific P2X7R inhibitor A 438079 (HY-15488, MedChemExpress, Shanghai, China) was dissolved in normal saline and delivered via an intraperitoneal injection to PPT1 KI mice with seizures (30 mg/kg, twice a day for two days). Seizure behaviors were recorded by monitoring for 24 h after drug application.

### 4.8. IHC and Confocal Imaging

Mouse brains were fixed in 4.0% paraformaldehyde at room temperature after perfusion with cold 1 × phosphate-buffered saline (PBS). The brains were sectioned coronally. Before staining, the slices were incubated in a citrate buffer (pH 6) 25 min at 95 °C for antigen retrieval and then permeabilized with 0.1% Triton X-100 in PBS. After blocking with 10% normal goat serum, the slices were incubated with primary antibodies overnight at 4 °C, followed by incubation with Alexa Fluor-conjugated secondary antibodies for 1 h at room temperature in dark. The primary antibody used was Iba-1 (GB13105-1, 1:100, Servicebio), and the secondary antibody used was Cy3 conjugated Goat Anti-Rabbit IgG (H + L) (GB21303, 1:100, Servicebio). Cells were mounted using 4’-6-diamidino-2-phenylindole-Fluoromount G (D1306, Thermo Fisher), and fluorescence was visualized using a Leica SP8x Inverted Meta confocal microscope (Leica, Wetzlar, Hesse, Germany). Images were processed using LSM image browser software (University of Arizona, Tucson, AZ, USA). One slice (5 μm) per mouse in classical hippocampal coronal section (Interaural 1.86 mm, Bregma -1.94 mm) was selected for microglial characterization in morphological and number. The cell numbers of bilateral hippocampus in 3 PPT1 KI mice and 3 WT mice were counted using case viewer 2.0 (3DHISTECH Ltd. Budapest, Hungary) and ImageJ 1.52d (National Institutes of Health, Bethesda, MD, USA) at 400× multiples in the hippocampal CA1 and CA3 regions.

### 4.9. DAB-Staining IHC

Coronal sections of mouse brains from 3-, 5- and 7-month-old PPT1 KI mice and 7-month-old WT mice were used. All chemicals and partial service were provided by Wuhan Servicebio Technology CO., LTD. The sections were washed in TBS, and the antigen was retrieved in citric acid buffer for 20 min at 70 °C. Then, all brain sections were rinsed in TBS for 8 min before and after incubation in 10%MeOH and 3%H_2_O_2_ in TBS for 10 min. Sections were then incubated with primary anti-NeuN (diluted 1:500, GB13138-1, Servicebio, Wuhan, China) over night at 4 °C, after incubation in blocking reagent. Sections were rinsed in TBS for 8 min followed by incubation in the secondary antibody (HRP-conjugated Goat Anti-Mouse IgG (H + L), Servicebio, Wuhan, China) diluted 1:400 in Supermix for 1 h. Sections were then incubated in 0.05% DAB (3,30-diaminobenzidine) tetrahydrochloride. Slides were mounted for histology with micro cover slides. Sections were analyzed using case viewer 2.0 (3DHISTECH Ltd. Budapest, Hungary) and ImageJ 1.52d (National Institutes of Health, Bethesda, MD, USA) at 200× multiples in the hippocampal CA1.

### 4.10. Data Analysis

Sample size test was performed using MSST (Version 6.0.6, Medsci sample size tools, Medsci Co. Ltd., Beijing, China). All data statistics were performed using IBM SPSS Statistics 20.0.0 software (IBM, Armonk, NY, USA), R 3.6.2 (Bell Laboratories) and GraphPad Prism 6.02 (San Diego, CA, USA). Comparisons between the two groups were analyzed using the pair-wise *t* test and permutation *t* test. One-way analysis of variance (ANOVA) was used for comparison for three or more groups of data. Significance was set at *p* < 0.05 for all analyses.

## Figures and Tables

**Figure 1 ijms-23-05586-f001:**
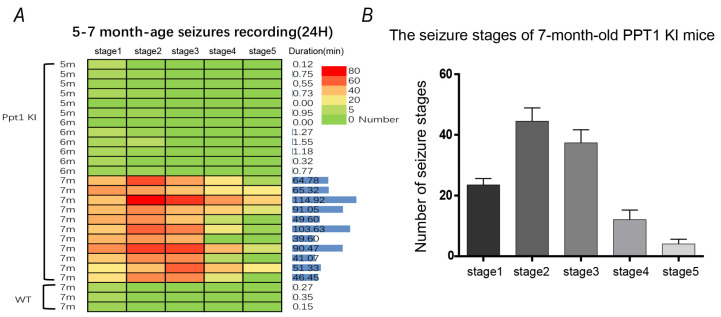
Seizure recordings in PPT1 KI mice (5–7 months old) and WT mice (7 months old) for 24 h. (**A**) Heatmap of seizures in PPT1 KI mice and WT mice for 24 h (PPT1 KI mice, *n* = 23, WT mice, *n* = 3); (**B**) Bar showing the number of different seizure stages in Racine scales of 7-month-old PPT1 KI mice for 24 h.

**Figure 2 ijms-23-05586-f002:**
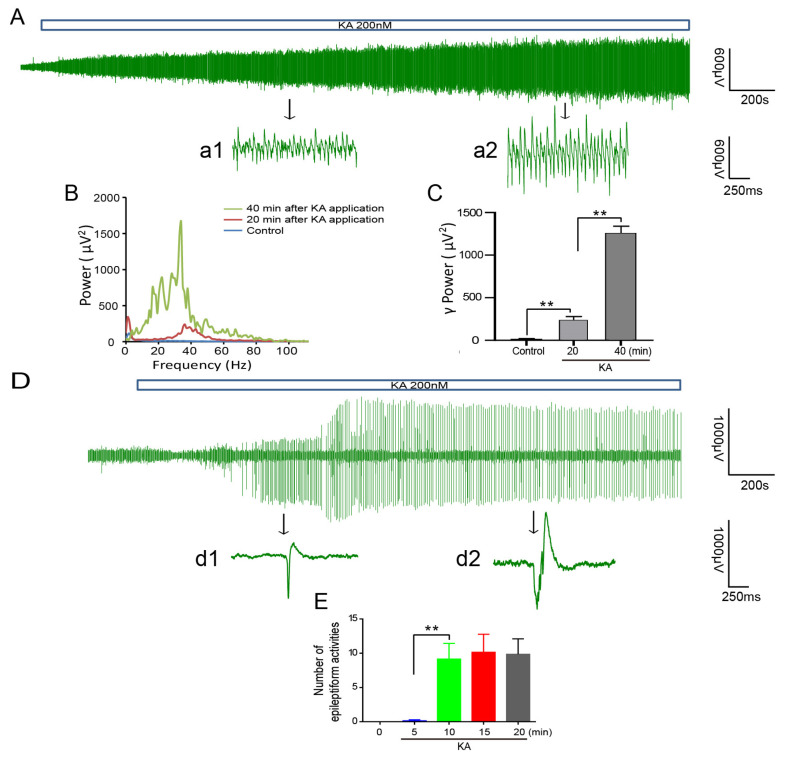
The representative local field potentials (LFPs) induced by 200 nM KA recorded in the CA3 region of WT mouse and PPT1 KI mouse. (**A**) The LFPs recordings for 50 min in a 7-month-old WT mouse (**top**). The representative 1 s LFPs recordings at 20 (a1) and 40 min (a2) after KA application (**bottom**). (**B**) The power spectrum shows that the power of oscillatory events. The blue, red and green lines in the chart represent the power spectra before KA application (control) at 20 and 40 min after KA application in a WT mouse. (**C**) Bar graph showing the γ power (the summated powers for the events of oscillatory frequency ranging from 20 to 60 Hz) of WT mice before KA, at 20 and 40 min after KA (ctrl vs. 20 min, *p* < 0.01, 20 vs. 40 min, *p* < 0.01, one way ANOVA on ranks following by post hoc LSD test, *n* = 16). (**D**) The LFPs recordings in a 7-month-old PPT1 KI mouse for 27 min (0.16–0.25 Hz) (**top**). The representative 1 s LFPs recordings at 8 (d1) and 20 min (d2) after KA application (**bottom**). (**E**) The number of epileptiform activities of PPT1 KI mice in different time points after KA (5 vs. 10 min, *p* < 0.01, one way ANOVA on ranks followed by post hoc LSD test, *n* = 17). ** *p* < 0.01.

**Figure 3 ijms-23-05586-f003:**
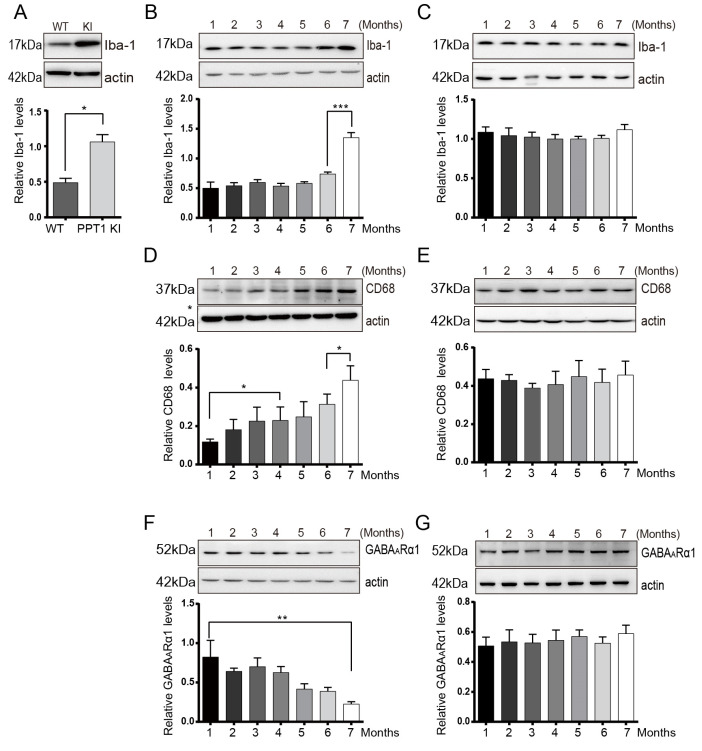
Western blot analysis of glial cell markers and synaptic receptors in hippocampus of PPT1 KI mice and WT mice. (**A**) Representative Western blots (**top**) and the bar graph (**bottom**) show the expression level of Iba-1 in hippocampus of 7-month-old PPT1 KI mice with seizures and age-matched WT mice (*p* = 0.04, Permutation *t* test, *n* = 6). (**B**,**C**) Representative Western blots (**top**) and the bar graph (**bottom**) show that expressions of Iba-1 in the hippocampus of PPT1 KI mice and age-matched WT mice from age 1 to 7 months (**B**, Iba-1 in PPT1 KI mice, 6 vs. 7 months old, *p* < 0.001, *n* = 35; **C**, Iba-1 in WT mice, *n* = 35); (**D**,**E**) Representative Western blots (**top**) and the bar graph (**bottom**) show expression of CD68 in the hippocampus of PPT1 KI mice and age-matched WT mice from age 1 to 7 months (**D**, CD68 in PPT1 KI mice, 1 vs. 4 months old, *p* = 0.049, 6 vs. 7 months old, *p* = 0.03, *n* = 21; **E**, CD68 in WT mice, *n* = 35); (**F**,**G**) Representative Western blots (**top**) and the bar graph (**bottom**) show expression of GABA_A_Rα1 in the hippocampus of PPT1 KI mice and age-matched WT mice from age 1 to 7 months (**F**, GABA_A_Rα1 in PPT1 KI mice, 1 vs. 7 months old, *p* < 0.01, *n* = 35; G, GABA_A_Rα1 in WT mice, *n* = 35). * *p* < 0.05; ** *p* < 0.01; ****p* < 0.001.

**Figure 5 ijms-23-05586-f005:**
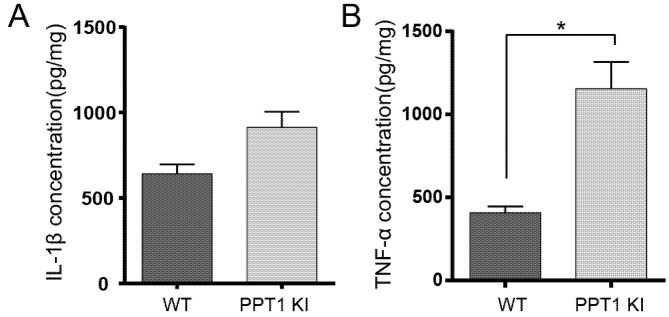
Changes in inflammatory cytokines IL-1β and TNF-α in PPT1 KI mice. (**A**) The expression level of IL-1β in the hippocampus from PPT1 KI mice and WT mice. (**B**) TNF-α protein in hippocampal tissue measured by ELISA assay. * *p* < 0.05.

**Figure 6 ijms-23-05586-f006:**
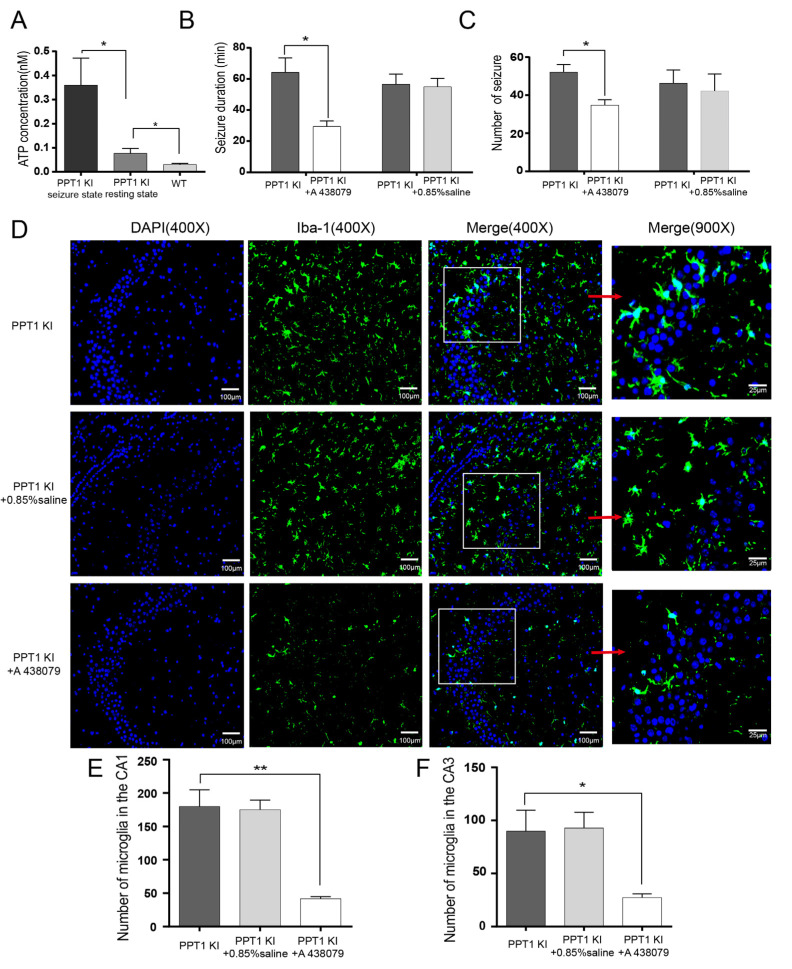
ATP concentration and seizure treatment in PPT1 KI mice. (**A**) ATP concentration in the hippocampus of 7-month-old PPT1 KI mice (seizure state and resting state) and age-matched WT mice (PPT1KI mice: seizure state vs. resting state, *p* = 0.043; PPT1KI mice at resting state vs. WT mice, *p* = 0.048, permutation *t* test, *n* = 9). (**B**) The seizure duration of PPT1 KI mice treated with A 438079 (*p* = 0.01, permutation *t* test, *n* = 8) and 0.85% saline (*n* = 4); (**C**) The number of seizure in PPT1 KI mice treated with A 438079 (*p* = 0.01, permutation *t* test, *n* = 8) and 0.85% saline (*n* = 4). (**D**) Representative images of confocal imaging in hippocampal CA3 area in 7-month-old PPT1 KI mice, 0.85% saline treatment and A 438079 treatment at 400 and 900× magnifications (DAPI in blue, Iba-1 in green). (**E**,**F**) The number of microglia in the CA1 and CA3 area of 7-month-old PPT1 KI mice, 0.85% saline treatment and A 438079 treatment (PPT1 KI mice vs. PPT1 KI mice after A 438079, CA1, *p* = 0.004, CA3, *p* = 0.02, permutation *t* test, *n* = 12). * *p* < 0.05; ** *p* < 0.01.

**Figure 7 ijms-23-05586-f007:**
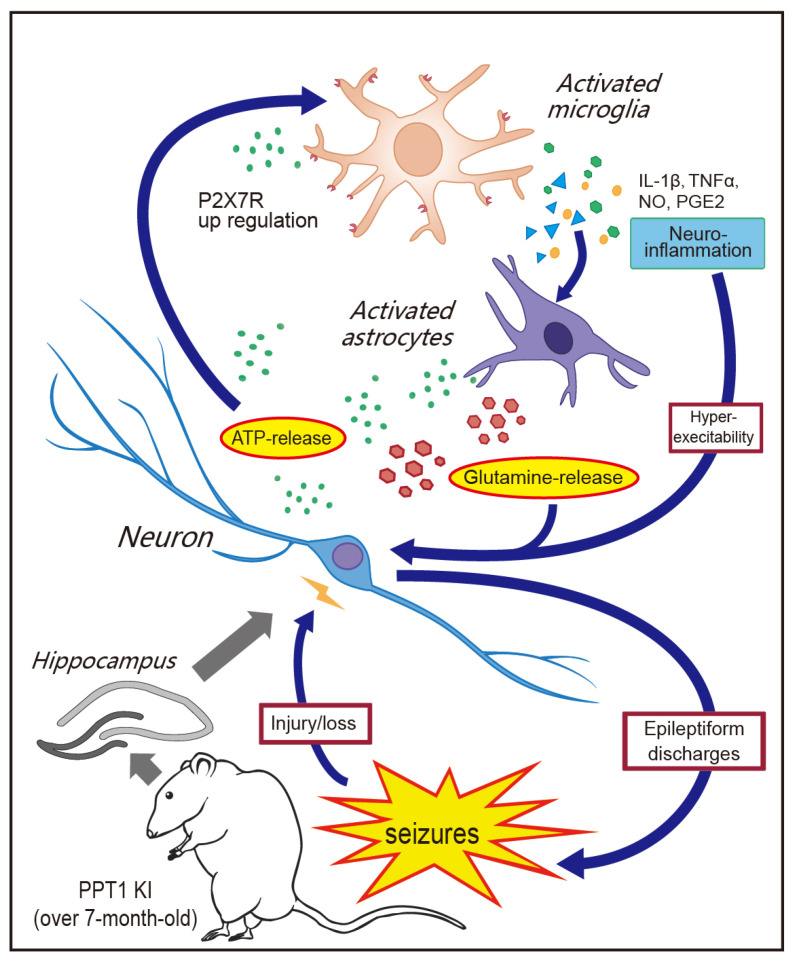
ATP-driven P2X7 activation in microglia likely contributes to seizures at a late age in PPT1 KI mice. Increased ATP concentration in the hippocampus of PPT1-deficient mice was observed in our study, which likely activate P2X7R on the surface of microglia, leading to inflammatory cytokines release. Inflammatory cytokines cause ATP and glutamate release from activated astrocytes, neuroinflammation, neuronal hyperexcitability and epileptiform discharges [34,54]. Neural damage and ATP release by activated microglia form a positive feedback loop in PPT1 KI mice.

## Data Availability

Data are contained within the article.

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
