# Peer review of "Seizures in PPT1 Knock-In Mice Are Associated with Inflammatory Activation of Microglia"

_ijms, 2022, doi:10.3390/ijms23105586_

Round 1

Reviewer 1 Report

This is an interesting paper with a good perspective on the purpose and aim of the study. However, I think that the Authors need to check and reconsider some of the following points:

1) In the results authors write that large and dense branches indicates that microglia cells are in activated state.

Section 2.6, line 199-201: “The microglia of the PPTI KI had large and dense branches, indicating these cells are highly activated”

I think his sentence should be reconsidered or rephrased. Although the morphological changes during microglial activation cannot be easily generalised because it is strongly depends on the type of the stimuli. Our overall and general knowledge about microglia activation is rather contradicting this statement made by the Authors.

Branching processes and larger dens process arborization are associated with ramified (resting) microglia. Upon activation microglia cells reduce their complexity and start to retract their processes which are resorbed into the cell body that results in an increased size of the somata. Also, high activation makes microglia cells turn into ameboid shape (Kettenmann et al, 2011; https://doi.org/10.1152/physrev.00011.2010 ). Avignone et al. (referenced by the Authors [35]) demonstrated the correlation between microglia morphology and the severity of seizures. Cell body size increased whereas the processes became smaller during activation caused by more intense seizure levels (Fig.1). Monif et al (also referenced by the Authors [27]) also demonstrated that microglia cells with branched primary processes rather display ramified morphology while activated cells show large somate with lamellipodias.

2) The results section should only include the details and statements of the experimental result. Theoretical background informations with references should be presented the introduction or the discussion section.

3) Grammar must be checked because some sentences do not make any sense or missing important things (e.g the object or the verb):

Some examples:

- Figure 2 (line 124-125):

 „The representative local field potentials (LFPs) induced by 200 nM recorded in the CA3 124 region of WT mouse and PPT1 KI mouse.” 

200 nM of what? Its kianic-acid (KA) but the authors missed to mention it in the sentence.

- Results (2.3, line 144-146):

„To further demonstrate the correlation between microglial activation and seizures in PPT1 KI mice, the number of microglia in the hippocampus of PPT1 KI mice of 1–7 months of age.” 

There is no verb in the second part of the sentence (e.g. was measured/calculated).

Also, the typos, the missing or extra spaces and the upper cases/lower cases within the sentences should be checked.

4) Authors should make it clear what kind of method was used to measure the cytokin levels. Based on the information in the materials section cytokin levels were deretmined by ELISA. However, in the results (2.8, line 237) authors write the following:

„We measured two common cytokines IL-1β and TNF-α by Western blot or ELISA”

5) Based on the materials section (4.5, line 140-141) ELISA results are expressed as pg/mg protein but in Figure 5 the results are expressed as pg/ml.

6) I also found some issues in the materals and methods section which should be checked/corrected or explained by the authors:

- Please indicate what kind of method was used to measure the total protein level of the tissue samples.

- Reference (catalog) number of the antibodies must be checked because some of them are not working and the products cannot be found in the manufacturers website:

  • No E-AB-40066 for the primary antibody CD68 used for western blot is not working
  • No GB131051 for the primary antibody Iba1 used for IHC is not working

Additional note: The Iba1 primary antibody used for western blot (A19776, ABclonal) is suitable for IHC according to the manufacturers website, so the authors would have used that Ab for the IHC as well. Please indicate the reason of using alternate Iba1 antibody for IHC.

- Section 4.5:

Please indicate the sensitivity and the intra - and interassay CV% of the ELISA tests.

- Section 4.8:

For antigen retrieval the samples were incubated overnight in citrate buffer at 95 oC. Overnight incubation in this temperature seems too harsh for me. Retrieval with citric buffer at this temperature usually lasts for 20-45 min (Krenacs et al. 10.1007/978-1-59745-324-0_14 ). Why was it necessary to  „boil” the 5 μm sections overnight? I am not sure whether this long incubation at this high temperature would not cause any significant tissue deterioration?

- Section 4.9:

Please indicate the time of the antigen retrieval (line 474).

Based on the manufacturers website the used anti-NeuN primary antibody (GB13138-1, Servicebio) is a mouse monoclonal AB. Staining mouse tissue with mouse antibody is a complicated process because off-target staining can cause high background. What was the Authors technique to avoid the unnecessary background staining?

Furthermore, Authors stated (line 478) that they used biotinylated goat-anti-rabbit secondary antibody which is not suitable for this staining if the the host of the primary NeuN antibody is mouse.

Additional note: In the applications sheet of the antibody provided by the manufacturer it is stated that the NeuN antibody is suitable for IHC on human samples only. Was there any validation performed by the Authors to demonstrate that the antibody is suitable for IHC on mouse samples?  

Author Response

Dear reviewer

    Thanks for your corrections and suggestions. We responsed your comments and modified our article.

Kind regards,
Mr. Zhang

Reviewer 2 Report

At the manuscript " Seizures in PPT1 knock-in mice are associated with inflammatory activation of microglia" by Dr. Xusheng Zhang et al authors described results of the study of the seizures’ activity and pathological changes in PPT1 knockout (PPT1 KI) mice. After performing impressive seria of experiments authors believe they confirmed that seizure in PPT1 KI mice may be associated with microglial activation involving in ATP-sensitive-P2X7R signaling and damaged inhibitory neurotransmission. The hypothesis of the authors is justified and I have no complaints about the experiments carried out.

Questions:

Do the authors have any information or hypothesis regarding whether protoplasmic or fibrous astrocytes are involved in the scheme proposed by the authors?

Is it possible to suppress the positive feedback loop that the authors write about? I mean neural damage and ATP release by activated microglia. Perhaps the reference are required here

Minor criticism:

  1. LINE 449: The specific P2X7R inhibitor A 438,079 (HY-15488, MedChemExpress, China) was… I think there is a typo here - an extra comma in the name of the reagent
  2. The scale bar in Figure 6D is completely unclear

The presentation of a subject is systematic and comprehensive, list of references is quite full and statistical analysis is proper. I am happy to recommend the manuscript for the publication after minor corrections mentioned above.

Author Response

At the manuscript " Seizures in PPT1 knock-in mice are associated with inflammatory activation of microglia" by Dr. Xusheng Zhang et al authors described results of the study of the seizures’ activity and pathological changes in PPT1 knockout (PPT1 KI) mice. After performing impressive seria of experiments authors believe they confirmed that seizure in PPT1 KI mice may be associated with microglial activation involving in ATP-sensitive-P2X7R signaling and damaged inhibitory neurotransmission. The hypothesis of the authors is justified and I have no complaints about the experiments carried out.

Questions:

Do the authors have any information or hypothesis regarding whether protoplasmic or fibrous astrocytes are involved in the scheme proposed by the authors?

Response: Thanks for your question. Astrocytes do play an important role in seizures(1-3). A Clinicopathologic Study reported that patients had chronic seizures with hyaline protoplasmic astrocytopathy(4), which suggest protoplasmic astrocytes may be associated with seizures, but there is insufficient evidence to support protoplasmic or fibrous astrocyte involvement in seizure.

Is it possible to suppress the positive feedback loop that the authors write about? I mean neural damage and ATP release by activated microglia. Perhaps the reference are required here

Response: it is possible to suppress the positive feedback loop by reducing or blocking ATP, proinflammatory cytokines release from damage neurons or activated astrocytes or microglia. For example, the inhibition of microglia activation using minocycline is a way to block the positive feedback loop of seizure(5-7).

Minor criticism:

  1. LINE 449: The specific P2X7R inhibitor A 438,079 (HY-15488, MedChemExpress, China) was… I think there is a typo here - an extra comma in the name of the reagent
  2. The scale bar in Figure 6D is completely unclear.

Response: Thanks for your correction, we modified these mistakes in our new version.

The presentation of a subject is systematic and comprehensive, list of references is quite full and statistical analysis is proper. I am happy to recommend the manuscript for the publication after minor corrections mentioned above.

Reference:

  1. F. Chan et al., The role of astrocytes in seizure generation: insights from a novel in vitro seizure model based on mitochondrial dysfunction. Brain 142, 391-411 (2019).
  2. Z. Setkowicz, E. Kosonowska, K. Janeczko, Inflammation in the developing rat modulates astroglial reactivity to seizures in the mature brain. J Anat 231, 366-379 (2017).
  3. G. F. Tian et al., An astrocytic basis of epilepsy. Nat Med 11, 973-981 (2005).
  4. R. A. Prayson, Hyaline Protoplasmic Astrocytopathy: A Clinicopathologic Study. Am J Clin Pathol 146, 503-509 (2016).
  5. U. B. Eyo et al., Neuronal hyperactivity recruits microglial processes via neuronal NMDA receptors and microglial P2Y12 receptors after status epilepticus. J Neurosci 34, 10528-10540 (2014).
  6. S. E. Haynes et al., The P2Y12 receptor regulates microglial activation by extracellular nucleotides. Nat Neurosci 9, 1512-1519 (2006).
  7. A. Badimon et al., Negative feedback control of neuronal activity by microglia. Nature 586, 417-423 (2020).

Reviewer 3 Report

I requested ALL blots, but never received them. This gives me the impression of having little confidence in the data presented in this manuscript.

Author Response

We're sorry you didn't receive our data,but we did send our all raw data of Westeren blots on 22 Apr 2022 14:22. We provided e-mail screenshot for you.

Round 2

Reviewer 1 Report

I appreciate that the Authors considered my advices and made some changes which was necessary to improve the manuscript.

However, the following points still need to be checked:

Methods section (Line 478-479):

The incubation time of the antigen retrieval at 70 oC is still missing.

ELISA results:

Authors must standardize the results.

In the first review I asked the Authors to clarify which unit were used in to express the cytokine level because in the graph axis pg/ml was the unit but in the sections pg/mg was used. The Authors changed the unit in the sections to pg/ml. So the results now expressed in pg/ml. But, if the samples are tissue homogenates (like in this study) the results should be expressed in pg/mg.

Because every tissue sample has different amount of total protein concentration the ELISA results (which are measured in pg/ml) must be standardized to the total protein concentrations and therefore expressed in pg/mg otherwise it is not possible to decide whether the difference between the cytokine level is caused by (in this case) the difference of the WT and PPT1 KI mice or just the different protein concentrations of each samples.

Small note:

There are small changes in the introduction section. Line 79-84 now is the following:

"Activated by ATP that released from neuronal terminal or leaked from the damaged cellular membrane of neurons, or outpoured from astrocytes[34], P2X7 mediates inflammatory activation of microglia. Activated microglia release inflammatory cytokines, which results in neuronal inflammation, damage, and seizure onset [35-38]."

For some reasion there is a crossed out section in the sentences but without those parts the sentences do not make sense.

Please check that paragraph.

Author Response

I appreciate that the Authors considered my advices and made some changes which was necessary to improve the manuscript.

However, the following points still need to be checked:

Methods section (Line 478-479):

The incubation time of the antigen retrieval at 70 oC is still missing.

Response:The incubation time of the antigen retrieval at 70 oC is 20 min, we added this information in the newly revised version.

ELISA results:

Authors must standardize the results.

In the first review I asked the Authors to clarify which unit were used in to express the cytokine level because in the graph axis pg/ml was the unit but in the sections pg/mg was used. The Authors changed the unit in the sections to pg/ml. So the results now expressed in pg/ml. But, if the samples are tissue homogenates (like in this study) the results should be expressed in pg/mg.

Because every tissue sample has different amount of total protein concentration the ELISA results (which are measured in pg/ml) must be standardized to the total protein concentrations and therefore expressed in pg/mg otherwise it is not possible to decide whether the difference between the cytokine level is caused by (in this case) the difference of the WT and PPT1 KI mice or just the different protein concentrations of each samples.

Response:Thank you for your suggestion. We have now standardized the ELISA results as expressed in pg/mg.

Small note:

There are small changes in the introduction section. Line 79-84 now is the following:

"Activated by ATP that released from neuronal terminal or leaked from the damaged cellular membrane of neurons, or outpoured from astrocytes[34], P2X7 mediates inflammatory activation of microglia. Activated microglia release inflammatory cytokines, which results in neuronal inflammation, damage, and seizure onset [35-38]."

For some reasion there is a crossed out section in the sentences but without those parts the sentences do not make sense.

.Please check that paragraph.

Response: We corrected this mistake as suggested.

Reviewer 3 Report

Thank you very much to the authors for sending full documentation of the experiments they carried out. The work is very well planned, done and written, and it exhaustively achieved the set goals.

I only have a few minor suggestions for improvement:

  1. Line 378-379: The authors write: All animal procedures complied with the animal study protocol (# 10-012) approved by the National Institutes of Health, Bethesda, Animal Care, USA). The research was done in China and financed by Chinese institutions and foundations, so why this strange sentence? Please correct it.

  1. Line 488: Please correct the manufacturer of the GraphPad Prism software (should be: GraphPad Software, San Diego, CA, USA).

  1. It would be reasonable to cite a recent review paper that also provides up-to-date information on purinergic receptors (doi: 10.1016 / j.pharmthera.2021.107821).

Author Response

Thank you very much to the authors for sending full documentation of the experiments they carried out. The work is very well planned, done and written, and it exhaustively achieved the set goals.

I only have a few minor suggestions for improvement:

  1. Line 378-379: The authors write: All animal procedures complied with the animal study protocol (# 10-012) approved by the National Institutes of Health, Bethesda, Animal Care, USA). The research was done in China and financed by Chinese institutions and foundations, so why this strange sentence? Please correct it.A

Response: Thanks for your suggestion and made correction in revised version.

  1. Line 488: Please correct the manufacturer of the GraphPad Prism software (should be: GraphPad Software, San Diego, CA, USA).

Response: Thanks for your suggestion, we corrected this error as suggested.

  1. It would be reasonable to cite a recent review paper that also provides up-to-date information on purinergic receptors (doi: 10.1016 / j.pharmthera.2021.107821).

    Response: We thank reviewer’s good suggestion, and cited the updated reference (doi: 10.1016 / j.pharmthera.2021.107821) in our article (line 533-534).

Round 3

Reviewer 1 Report

I appreciate that the Authors corrected the results and standardized the cytokine levels which I think improved the overall look of the graphs in figure 5 as well.

Author Response

We appreciate your help and suggestions in the article.